# Chemokine Receptors CCR6 and PD1 Blocking scFv E27 Enhances Anti-EGFR CAR-T Therapeutic Efficacy in a Preclinical Model of Human Non-Small Cell Lung Carcinoma

**DOI:** 10.3390/ijms24065424

**Published:** 2023-03-12

**Authors:** Jing Wang, Yanan Wang, Hanyu Pan, Lin Zhao, Xinyi Yang, Zhiming Liang, Xiaoting Shen, Jing Zhang, Jinlong Yang, Yuqi Zhu, Jingna Xun, Yue Liang, Qinru Lin, Huitong Liang, Min Li, Huanzhang Zhu

**Affiliations:** 1State Key Laboratory of Genetic Engineering and Engineering Research Center of Gene Technology, Ministry of Education, Institute of Genetics, School of Life Sciences, Fudan University, Shanghai 200082, China; 2School of Life Sciences, Fudan University, Shanghai 200082, China

**Keywords:** CAR T cell therapy, EGFR, NSCLC, chemokine, checkpoint, migration, CCR6, PD1

## Abstract

Chimeric antigen receptor (CAR)-T cells, a therapeutic agent for solid tumors, are not completely effective due to a lack of infiltration of T cells into the tumor site and immunity caused by Programmed Death Receptor 1(PD1). Here, an epidermal growth factor receptor (EGFR) CAR-T cell was engineered to express the chemokine receptor CCR6 and secrete PD1 blocking Single-chain antibody fragment (scFv) E27 to enhance their anti-tumor effects. The findings showed that CCR6 enhanced the migration of EGFR CAR-E27-CCR6 T cells in vitro by the Transwell migration assay. When incubated with tumor cells, EGFR CAR-E27-CCR6 T cells specifically exerted potent cytotoxicity and produced high levels of pro-inflammatory cytokines, including tumor necrosis factor-α (TNF-α), interleukin-2 (IL-2), and interferon-γ (IFN-γ). A non-small cell lung carcinoma (NSCLC) cell line-derived xenograft model was constructed by implanting modified A549 cell lines into immunodeficient NOD.*Prkdc^scid^Il2rg^em1^/Smoc* (NSG) mice. In comparison with traditional EGFR CAR-T cells, live imaging indicated that EGFR CAR-E27-CCR6 T cells displayed superior anti-tumor function. In addition, the histopathological examination of mouse organs showed no obvious organic damage. Our findings confirmed that PD1 blocking and CCR6 can enhance the anti-tumor function of EGFR CAR-T cells in an NSCLC xenograft model, providing an effective treatment strategy to improve the efficacy of CAR-T in NSCLC.

## 1. Introduction

According to the World Health Organization, in 2020 the incidence of lung cancer (12.2%) worldwide was second only to that of breast cancer, making it one of the major diseases that threaten human health and life. Notably, lung cancer was found to be the most lethal cancer, accounting for 18.2% of all cancer-related deaths [1].

Non-small cell lung carcinoma (NSCLC) accounts for approximately 85% of all lung cancers [2], and has a dismal 5-year survival rate of 5% [3]. Surgery, radiotherapy, and chemotherapy are the main treatment methods for NSCLC. However, their curative effect is limited, and consequently the ordinary stage (IIIA) 5-year survival rate is a meagre 15% [4]. In the past 20 years, lung cancer therapies have broadened and innovated, with tumor immunotherapy [5,6] being at the forefront. In particular, chimeric antigen receptor (CAR)-T therapy has received attention because of its higher safety and efficacy [7], where it directly recognizes tumor antigens in a major histocompatibility complex (MHC)-independent manner [8]. CARs are engineered synthetic receptors that are incorporated into the cell membranes of lymphocytes, most commonly T cells, redirecting them to recognize and eliminate cells expressing a specific target antigen. CARs function by binding to target antigens expressed on the cell surface in a process that is independent from that of MHC receptors, thereby stimulating vigorous T cell activation leading to a powerful anti-tumor response [9]. CAR-T therapy has also been proven useful in the treatment of hematologic tumors [10]. Multiple CAR-T cell products were approved by the Food and Drug Administration (FDA) by the year 2022 [11,12,13]. Their use in solid tumor research is still ongoing [14], but many preclinical studies have demonstrated superior efficacy in the treatment of solid tumor models, including glioblastoma [15], sarcoma [16], neuroblastoma [17], breast cancer [18], advanced gastric or pancreatic cancer [19], lung cancer [20], hepatocellular carcinoma [21], ovarian cancer [22], and prostate cancer [23]. Clinical trials of CAR-T therapy targeting mesothelin (MSLN) [24], epidermal growth factor receptor (EGFR) [25], human epidermal growth factor receptor 2 (HER2) [26], and carcinoembryonic antigen (CEA) [27] in NSCLC have been completed in recent years, but with disappointing results.

EGFR is a commonly used molecular target in CAR-T treatment of NSCLC [28]. EGFR belongs to the ERBB (receptor protein-tyrosine kinase) family of tyrosine kinase receptors. The EGFR signaling cascade is a key regulator of cell proliferation, differentiation, division, survival, and cancer development [29]. As of February 2022, four clinical trials have used EGFR-targeted CAR-T cells to treat NSCLC (NCT04153799, NCT05060796, NCT04592666, and NCT04025216). More than 60% of NSCLC cases have been reported previously with highly expressed EGFR [30,31]. The expression level of EGFR is closely related to clinical prognosis [32]; therefore, it may be a promising target for tumor immunotherapy in NSCLC. Inhibition of the tumor microenvironment (TME) [33] is one of the main obstacles in the treatment of solid tumors. Epitopes related to immune checkpoints on the surface of certain cancer cells can inhibit immune system functioning [34]; therefore, research on immune checkpoints is important in the field of cancer immunotherapy research [35,36,37]. Programmed cell death protein 1 (PD1 and its ligand PDL1) and cytotoxic T lymphocyte-associated antigen-4 (CTLA-4) are the main immune checkpoints involved in cancer therapy [38,39,40,41].

Another cause of unsatisfactory CAR-T cell treatment for solid tumors may be the limited migration of CAR-T cells. An efficient migration to and infiltration into tumor tissue by CAR-T cells is required to effectively kill tumor cells and eliminate tumors [42]. The potential of CAR-T cell migration may be enhanced by the influence of chemokines on leukocyte recruitment, angiogenesis, tumor growth, proliferation, and metastasis [43]. Chemokines CCL19, CXCL13, and CCL20 are chemokines with high expression [44,45] in lung adenocarcinoma, a conclusion of The Cancer Genome Atlas (TCGA) project initiated by the National Cancer Institute in the United States. Therefore, the expression of CCR6, the receptor of CCL20, may promote the migration of immune cells to solid tumor sites, especially during CAR-T therapy. Recently, there have been many studies on CAR-T application when combined with checkpoint blockade [46] or chemokines [47,48] to improve anti-tumor efficacy. Here, we engineered an EGFR CAR-T cell to express chemokine receptor CCR6 and secrete PD1 blocking scFv E27 (a single-chain fragment variable of an antibody against PD-1 named E27), and explored whether the PD1 blockade and chemokine receptor CCR6 can promote CAR-T cell anti-tumor activity in vitro and in xenograft tumor models.

## 2. Results

### 2.1. Acquisition and Characterization of Human Primary T Cells Expressing EGFR CAR, PD1 Blocking scFv (E27) and CCR6

To obtain EGFR CAR-T, EGFR CAR-E27-T, and EGFR CAR-E27-CCR6-T cells, the DNA sequence of anti-EGFR scFv, E27 and CCR6 was collected from former articles [49], and lentiviral vectors for EGFR CAR, EGFR CAR-E27, and EGFR CAR-E27-CCR6 were constructed as described in Section 4 (Figure 1A). The cells from CD3+T were isolated from human peripheral blood mononuclear cells (PBMCs) of healthy donors using an isolation kit and then infected with lentivirus after 72 h of activation. The activation, phenotype, and expression of CAR- and CCR6-T cells were detected, and the results showed that 24.1, 49.3, and 68.3% of them (Figure 1B) expressed anti-EGFR scFv on EGFR CAR-T, EGFR CAR-E27-T and EGFR CAR-E27-CCR6-T cells, and 33.2% expressed CCR6 in EGFR-CAR-E27-CCR6-T cells (Figure 1C). The expression and secretion of E27 were detected by western blotting, and the results showed that both EGFR CAR-E27-T and EGFR CAR-E27-CCR6-T cells expressed and secreted E27 in the supernatant and intracellular fluid (Figure 1D).

The initial objective of this study was to determine the proliferation rate of MOCK T, EGFR CAR-T, EGFR CAR-E27-T, and EGFR CAR-E27-CCR6-T cells after activation and lentiviral infection. The proliferation ability of T cells can be enhanced by modifying CAR and E27, as shown in the results. However, the expression of CCR6 did not significantly affect the CAR-T cell proliferation significantly (Figure 1E).

To identify the function of E27, PD1 expression was tested to identify the function of E27 at 72 h after the lentiviral infection of T cells. Compared with MOCK T cells, the expression of PD1 was almost undetectable in both EGFR CAR-E27-T and EGFR CAR-E27-CCR6-T cells, confirming the blocking effect of E27 on PD1 (Figure 1F). This may be due to the single-chain antibody E27 secreted by CAR-T which binds to the PD1 receptor on the CAR-T surface, meaning that the secretion of E27 has functional activity.

Most T cells were in the early and middle stages of activation (Figure 2A). The expression levels of CD69 and CD25 on the surface of activated T cells were 25 and 71%, respectively, after 48 h of activation. As the phenotype of T cells affects their killing efficiency, we measured the phenotype of each CAR-T cell. The proportion of CD45RA and CD62L double-positive cells (naive T cells) in the CAR-T cells was about 20%, and the anti-tumor effect of T cells (Figure 2B) was beneficial, as the proportion of CD62L single-positive cells (memory T cells) was 80%. The ratio of CD4+ to CD8+ was tested for the efficiency of T cell separation, where 33% was CD8 single-positive and 46% (Figure 2C) was CD4 single-positive. The T-cell results in the above groups were similar.

### 2.2. Establishment of NSCLC Cell Lines Expressing EGFR, Luc, PDL1 and CCL20

The main reason for establishing target cells in two steps was to obtain A549 cell lines with high expression of Luc, PDL1, and CCL20. A549 cells were transduced with pCDH-PDL1-Luc-puro lentivirus, followed by puromycin selection. A549-PDL1-Luc cells were transduced with FUGW-CCL20-Hygro lentivirus and selected with Hygromycin B (Figure 3A). The expression of EGFR in the NSCLC cell line A549 was confirmed using flow cytometry. The EGFR expression in A549 cells was 64% (Figure 3B). EGFR is not overexpressed in A549, in order to make it familiar with the condition of the NSCLC patients (EGFR was expressed in 40–80% of patient tumor tissue [32]). The expression of PDL1 on the surface of A549 cells after stable transfection was 90.7% (Figure 3C), as shown by the FACS analysis. The luciferase expression of luciferase was used for cytotoxicity experiments and live imaging (Figure 3D). The reason the expression of CCL20 in stably transfected A549-PDL1-Luc-CCL20 cells is statistically different is that it is compared with the uninfected cell line and the A549 cell line that only express PDL1 and Luc. The PDL1+/Luc+/CCL20+ target cell line (Figure 3E) was successfully established.

### 2.3. Effect of CCR6 on EGFR CAR-T Cell Migration In Vitro

To validate the in vitro chemotaxis ability of the CCR6-CCL20 axis, we used the Transwell migration assay, which showed that T cells could migrate through the 5 µm polycarbonate membrane to the lower chamber containing the supernatant of A549-PDL1-Luc-CCL20 cells (Figure 3F). The migration of CAR-T cells expressing CCR6 is significantly different from other groups, just over three times that of the MOCK T group, which is what the results show. There was no significant difference among the EGFR CAR-T, EGFR CAR-E27-T, and MOCK T groups; therefore, CCR6 was functional in response to CCL20 secreted by tumor cells in vitro (Figure 3G).

### 2.4. EGFR CAR and E27 Improved the Cytotoxicity Effects of T Cells In Vitro

To explore the antitumor activity of different EGFR CAR-T groups in vitro, we co-incubated the immune and target cells at ratios of 5:1, 10:1, and 20:1—and the killing effects were assessed by detecting the decrease in the fluorescence value of target cells. MOCK T, EGFR CAR-T, EGFR CAR-E27-T, and EGFR CAR-E27-CCR6-T cells specifically lysed A549-PDL1-Luc-CCL20 cells at different effector-target (E/T) ratios. The modification of EGFR CAR and E27 significantly improved the cytotoxicity of T cells compared with MOCK T cells. The modification of CCR6 did not increase the cytolytic activity of EGFR-CAR T cells in vitro. When the E/T ratio was 20:1, the result was 74.34%, and the killing efficiency of CAR-T cells to target cells was 6 h after co-incubation, which was nearly three times that of the MOCK T group (Figure 4A). After co-incubation of MOCK T, EGFR CAR-T, EGFR CAR-E27-T, and EGFR CAR-E27-CCR6-T with target cells A549-PDL1-Luc-CCL20 at an E/T ratio of 10:1 for 8 h, different levels of IL-2, IFN-γ, and TNF-α were produced. The secretion of inflammatory cytokines in the EGFR CAR and E27 groups was significantly higher than that in the other groups, but the data between the CCR6 and CCR6-free groups showed little difference (Figure 4B–D).

### 2.5. Anti-Tumor Activity of EGFR CAR-E27-CCR6-T Cells in a NSCLC Xenograft Model

The antitumor effect on target cells in vivo was further verified by EGFR CAR-T, EGFR CAR-E27-T, and EGFR CAR-E27-CCR6-T, and the tumor transplantation model of NSG mice was established (Figure 5A). Seven days after the tumor was implanted, all mice groups had obvious fluorescence imaging at the tumor sites (~10,000 cps). Subsequently, EGFR CAR-T, EGFR CAR-E27-T, EGFR CAR-E27-CCR6-T, and MOCK T cells were injected into the mice via the tail vein on days 7, 10, and 13. Weight measurements, fluorescent imaging in vivo, and health monitoring were performed every seven days.

By day 32, the bioluminescence value of all mice in the MOCK T group exceeded the threshold (65,535 cps), while that of the other three groups was much lower. The tumor reduction rates in the EGFR CAR-T and EGFR CAR-E27-T groups were 40% and 80%, respectively, and those in the EGFR CAR-E27-CCR6-T group were almost completely eliminated (Figure 5B). By day 39, 100%, 80%, 40%, and 20% of mice survived in the EGFR CAR-E27-CCR6-T, EGFR CAR-E27-T, EGFR CAR-T, and MOCK T groups, respectively (*n* = 5, Figure 5B).

By counting the highest bioluminescence values at each time point, the tumor differences, such as size, among the four mice groups could be seen more clearly (Figure 5C). The tumor tissue weights showed statistical differences between the MOCK T group and the other groups (Figure 5E). In addition, dissection, photography, and quantification were performed on all mouse tumors. The tumor tissues showed statistical differences between the MOCK T group and the other groups (Figure 5D).

### 2.6. Persistence, Penetration, and Safety of CAR-T Cells in Various Tissues of NSG Mice

The CAR-T cell presence was tested in the blood and organs of the mice. Blood was drawn (100 μL) three weeks after the NSG mice were implanted with CAR-T cells, and the lung, heart, liver, spleen, and kidney organs were dissected when all mice died. A polymerase chain reaction (PCR) was performed on genomic DNA extracted from the blood and organs using specific primers for DNA encoding anti-EGFR scFv. The results showed that specific 476 bp DNA bands were only detected in the blood (Figure 6a) and organs (Figure 6b) of the CAR-T group mice, verifying the survival and expansion of CAR-T cells here.

The spleen was selected as the representative organ for detailed analysis. We dissected the mice to obtain spleen tissue and stained it with a human CD3 immunohistochemical antibody. Immunohistochemical results showed that there were almost no CD3+ cells in the spleen tissue of the MOCK T group, while there were a certain number in the EGFR CAR-T, EGFR CAR-E27-T, and EGFR CAR-E27-CCR6-T groups (Figure 6c), and the difference was statistically significant (Figure 6d).

EGFR CAR, E27, and CCR6 promote the homing of immune cells, indicating better antitumor effects of EGFR CAR-E27-CCR6-T than other T cells. H&E staining was also performed on other organs of the mice in each group, and the safety of CAR-T cells was preliminarily evaluated based on cell morphology and arrangement (Appendix A).

## 3. Discussion

Currently, CAR-T therapy shows safer and better curative effects in the treatment of hematological malignancies than conventional therapies for many types of cancer. However, the efficacy of CAR-T cells in treating solid tumors is limited [50].

To overcome the limitations of solid tumor treatment and to explore safer and more effective CARs for the treatment of NSCLC, we designed a modified EGFR CAR-E27-CCR6, which can secrete PD1 single-chain scFv E27 and express the chemokine receptor CCR6. Here, we found that (i) the anti-EGFR single-chain antibody, PD1 single-chain antibody (E27), and CCR6 were stably expressed on the surface of EGFR-CAR T cells; (ii) these EGFR CAR T cells possessed a higher proportion of central memory T cells and initial T cells with higher cytotoxicity; and (iii) they possessed in vitro- and in vivo-specific and potent cytotoxicity against NSCLC cells, especially EGFR CAR-E27-CCR6-T. Our study revealed the therapeutic potential of our modified CAR-T cells against EGFR-positive human cancers such as NSCLC.

To find potential therapeutic strategies to overcome the obstacles in the treatment of solid tumors, we aimed to improve the anticancer effect and persistence of CAR-T cells. The combination of CAR-T with other immunotherapies is a potentially promising strategy; an example being immune checkpoint inhibitor (ICI) therapy and the chemokine receptor-based optimization of T-cell trafficking. Moreover, single-chain antibodies to ICIs have shown strong antitumor effects in some studies targeting CAR-T therapy for solid tumors, with ICIs proving to be effective against various cancer types. The in situ secretion of anti-PD-1 antibodies by CAR-T cells may limit systemic antibody exposure, thus reducing toxicity. This approach has been shown to enhance the preclinical antitumor efficacy of CAR-T cells in various hematologic and solid tumors [51]. Zhou et al. applied PD1 scFv and EGFR CAR-T cell therapy together to eliminate gastric cancer in 2020 [52], showing better efficacy than with single CAR-T cells. The design of T cells modified with both CARs engineered with checkpoint blocking could allow better T cell activation in tumor lesions and the effective reduction of systemic toxicity. The use of chemokine receptors together with CAR-T therapy has also proven to be a good strategy against solid tumors. Jin et al. verified the effect of CXCR1 and CXCR2 in enhancing the migration and persistence of CAR-T cells in glioblastoma, ovarian cancer, and pancreatic cancer [53].

Furthermore, our study showed that EGFR-CAR CAR-T cell therapy was safe and had no off-target effects in mice. Because of the expression of EGFR in non-cancerous tumors, tolerable and controllable target-related toxicities, such as skin rash and diarrhea, have been seen on NSCLC patients in clinical trials [54,55]. We histopathologically examined the potential systemic toxicity in several major mouse organs and found that EGFR CAR-T cells did not cause any notable pathological damage (Appendix A).

This study has some limitations that need to be acknowledged. First, we overexpressed CCL20 in A549 lung cells to obtain more intuitive results. However, abundant CCL20 is not found on the surface of tumor tissues of all NSCLC patients, although it has been confirmed that it is highly expressed in adenocarcinoma tissues. The expression of CCL20 needs to be examined before clinical trials of this type of modified CAR therapy. Second, a mouse model was established by subcutaneously implanting tumor cells on the back waist of mice instead of in situ tumor implantation. No organic pathological damage has been observed in major organs, including the lungs, which makes this method far inferior to the real NSCLC condition in patients in our current study.

Taken together, the data obtained revealed that our modified EGFR CAR-T cells may serve as a highly effective treatment for NSCLC and have potential applications in its treatment. This type of therapy may also be a potential strategy for treating solid malignancies with EGFR overexpression.

## 4. Materials and Methods

### 4.1. Cell Lines and Human Primary T Cells

Cells were cultured in Dulbecco’s Modified Eagle’s Medium (DMEM), and the NSCLC cell line A549 was obtained from the cell bank of the National Science and Technology Infrastructure. 293FT-17 was maintained in DMEM. Jurkat cells cultured in RPMI 1640-media (Hyclone, Logan, UT, USA) were obtained from the American Type Culture Collection (ATCC). Fetal bovine serum (10% FBS), 100 U/mL penicillin, and 100 g/mL streptomycin (Gibco, Carlsbad, CA, USA) were added to all of the media. All cell lines were incubated in 5% CO_2_ at 37 °C. Lentiviral vector 1 expressing PDL1, luciferase, and puromycin resistance genes and lentiviral vector 2 expressing CCL20 and hygromycin B resistance genes, were transduced into A549 cells. After 14 days, 10 g/mL puromycin for 7 days and then 200 g/mL hygromycin B for 7 days were sequentially screened. The cell line A549-PDL1-Luc-CCL20 stably overexpressing PDL1, and CCL20 was obtained by culturing and amplification.

Peripheral blood samples from healthy donors were obtained from the Changhai Hospital of Shanghai, China. The PBMCs was separated using FicollPaque Premium (Cytiva, Global) by gradient centrifugation. The enrichment of CD3 T cells by negative selection from PBMC was performed using a Pan T cell isolation kit (Miltenyi Biotec, North Rhine-Westphalia, DE, Germany). CD3 T cells stimulated with 25 μL/mL immunocult human CD3/CD28 T cell activator (Stemcell, Los Angeles, CA, USA), 5 ng/mL rIL-2 (R&D Systems, Minneapolis, MN, USA), and 5 ng/mL rIL-15 (R&D Systems, MN, USA) were cultured in an ImmunoCult- xf T cell expansion medium (Stemcell, Los Angeles, CA, USA).

### 4.2. Construction of Plasmids

The amino acid sequence of the anti-EGFR single-chain antibody was obtained from the patent US8580263B2 of Andreas Lehmann et al. [49], and the DNA sequence was optimized and synthesized by Genewiz (Suzhou, China). The EGFR CAR contained a CD8α signal peptide (GenBank NM001768.6, 1–63 bp), anti-EGFR scFv, CD8α hinge, intracellular signaling domains of 4-1BB (GenBank NM001561.5, 640–765 bp), a transmembrane region (GenBank NM001768.6, 412–609 bp), and CD3ζ (GenBank NM 198253.2, 154–492 bp). EGFR CAR was amplified and cloned into the lentiviral backbone pTRPE, EGFR-CAR, and PD1 scFv (E27), and CCR6 was linked to the 2A peptide.

The DNA sequence of PD1 scFv (E27) was obtained from patent WO 2016/210129 A1 of Brentjens Renier et al., and the DNA sequence of CCR6 was obtained from GenBank U95626.1, 96,642–97,676 bp. The DNA sequence of PDL1 was obtained from GenBank NC000009.12, the DNA sequence of CCL20 was obtained from GenBank AY398937.1, and all of the above sequences were synthesized by Genewiz (Suzhou, China). The PDL1 and luciferase genes were cloned into the lentiviral vector PCDH containing the puromycin resistance gene, and CCL20 was cloned into the lentiviral vector FUGW containing the hygromycin B resistance gene. These plasmids were confirmed by double enzyme digestion and DNA sequencing.

### 4.3. Lentivirus Manufacture and Infection

The 293FT-17 cell line was transfected with polyethyleneimine (PEI) transfection reagent, and the three-plasmid system of PMD2.G, psPAX2, and the target plasmid was used to manufacture the lentivirus at a ratio of 3.4:6.8:10. At 4 °C, the virus was collected twice from the cell supernatant by ultracentrifugation at 25,000 rpm for 2 h. A virus storage solution was added, and the supernatant was discarded, left at 4 °C overnight, resuspended, and stored at −80 °C.

The slow virus titer was calculated by flow cytometry, and 293FT-17 cells were infected by gradient dilution of a concentrated virus solution containing 8 µg/mL polystyrene. CD3 T cells were infected with 8 µg/mL polystyrene (MOI ≈ 5) lentivirus concentrate after CD3 T cells were activated for 72 h. CAR and CCR6 expression was detected using flow cytometry 72 h after infection.

### 4.4. Detection and Identification of Target Genes on Target and Effector Cells

Flow cytometry. Flow cytometry antibodies will collect 1 × 10^4^ to 1 × 10^6^ cells according to the total number of cells. The cells were kept in dark ice for 30 min and then washed twice with PBS. The data analyzed using FlowJo software (BD Bioscience, Eugene, OR, USA) were detected using a Beckman Coulter cytometer. The flow cytometry antibodies used in this experiment were as follows: APC-conjugated streptavidin, Human FITC-conjugated anti-CD25, Human PE-conjugated anti-CD69, Human FITC-conjugated anti-CD4, Human PE-conjugated anti-CD3, Human PE-conjugated anti-CD8), Human PE-conjugated anti-CCR6, Human PE-conjugated anti-CD45RA, Human FITC-conjugated anti-CD62L, Human PE-conjugated anti-CD274, Human PE-conjugated anti-EGFR, Human PE-conjugated anti-CD107α, Human PE-conjugated anti-CD279 (all from BD Bioscience, OR, USA), and Human PE-conjugated recombinant EGFR (ACRO, Washington, DC, USA).

Western blot. CD3 + T cells were collected 72 h after lentivirus infection (without changing the medium). The supernatant and cell particles stored at −20 °C were collected after centrifugation at 300× *g* for 10 min. After the lysate 5× passive lysis was added to the cell ball, the protease inhibitor was added at a ratio of 1:100, and the cell ball was completely resuspended. Centrifuging at 12,000× *g* rpm for 20 min at 4 °C involves gently rotating the cell mixture, melting the contents in ice for 30 min, and storing the collected supernatant at −20 °C. After protein quantification, the supernatant was added to a 25% loading buffer by volume after protein quantification, and heated for 10 min in a 95 °C water bath.

SDS-PAGE electrophoresis was performed at a constant voltage of 80 V for 120 min, and the transmembrane was operated under a constant current of 240 mA for 120 min to transfer all bands to the PVDF membrane. The mixture was swirled gently and incubated for 90 min by adding 0.5% skim milk powder to the PVDF membrane. The primary antibody was added at a ratio of 2:5000, incubated overnight on ice using a gentle vortex in the dark, and washed three times with PBST. The secondary antibody was added at a ratio of 2:5000, incubated for 90 min at room temperature using a gentle vortex in the dark, washed three times with PBST, soaked in chromogenic solution for 1 min, and then exposed to a western blotting instrument.

The following western blotting antibodies were used here: mouse anti-human β-actin, goat anti-mouse IgG-HRP, goat anti-rabbit IgG-HRP (all from China TransGen Biotech, Beijing, China), and rabbit anti-human HA (CST, Boston, MA, USA).

Luciferase assay. In an opaque 96-well test plate, 100 μL cells was placed at appropriate concentrations. After 100 μL luciferase reporter reagent was added, they were mixed evenly on a shaking table at room temperature in the dark and cultured in the dark for 5–20 min at room temperature. Luminescence was detected using a microplate reader.

### 4.5. Cell Proliferation Assay

To prepare the assay, 5 × 10^3^ (100 μL) MOCK T cells, EGFR CAR-T cells, EGFR CAR-E27-T cells, and EGFR CAR-E27-CCR6-T cells were added to a transparent 96-well plate, and 10 μL CCK-8 (Yeasen Biotech, Shanghai, China) was added and mixed by vortexing. The cells were cultured at 37 °C and 5% CO_2_ for 2–6 h. After 24, 48, and 72 h, the absorbance value was measured using a microplate reader at a wavelength of 450 nm to compare the proliferation ability of CAR-T cells in different groups.

### 4.6. Transwell Assay

The Transwell assay was used to examine the effect of CCR6 on the migration efficiency of CAR-T cells. In the upper cavity of culture plates(Corning, New York, NY, USA) (100 μL) MOCK T, EGFR CAR-T, EGFR CAR-E27-T, and EGFR CAR-E27-CCR6-T cells with a diameter of 6.5 mm and a pore diameter of 5 μm were added. Simultaneously, A549-PDL1-Luc-CCL20 cell supernatant was cultured in the lower cavity of Corning for more than 48 h. The T cells migrated to the inferior cavity at 37 °C and were quantified by incubation with CCK-8 at 37 °C and 5% CO_2_ for 24 h.

### 4.7. Cytotoxicity and Cytokine Release Assays

Since the target cells carry the luciferase gene, luciferase detection was used to judge and compare the cytotoxicity of different groups of CAR-T cells against the target cells in vitro. The collected A549-PDL1-Luc-CCL20 cells were inoculated in a round-bottom 96-well plate and incubated for 4 h at 5 × 10^3^ cells/well (50 μL) at 37 °C and 5% CO_2_. MOCK T, EGFR CAR-T, EGFR CAR-E27-T, and EGFR CAR-E27-CCR6-T cells were collected and inoculated into a well plate with target cells at target-effect ratios of 1:5, 1:10, and 1:20. T cells of 2.5 × 10^4^, 5 × 10^4^, and 1 × 10^5^ (50 μL) were added and incubated at 37 °C for 6 h with 5% CO_2_. The culture solution was used as a control. The cell lysate (100 μL) was added to a 96-well plate (the supernatant was discarded), incubated at room temperature, and vortexed for 10 min. All resuspended material was transferred onto an opaque 96-well plate, 100 μL luciferase reporter reagent was added, and then incubated at room temperature in the dark for 5–30 min. The luminescence was detected using a microplate reader. The insecticidal quantity is the fluorescence value of the control group minus that of the experimental group, and insecticidal efficiency is the insecticidal quantity divided by the fluorescence value of the control group.

MOCK T, EGFR CAR-T, EGFR CAR-E27-T, EGFR CAR-E27-CCR6-T, and target cells A549-PDL1-Luc-CCL20 were co-incubated in a 12-well plate at a ratio of 10:1 for 8 h, and the supernatant was collected. ELISA precoating kits for IL-2, TNF-α, and IFN-ɤ (Dakewe Biotech, Shanghai, China) were used to detect the secretion levels of these three cytokines. Please refer to the product manual for the specific operating steps.

### 4.8. Establishment and In Vivo Experiments of a NSG Mouse Lung Cancer Tumor Model

Female NSG mice aged 4–8 weeks (NOD.*Prkdc^scid^Il2rg^em1^/Smoc*, NM-NSG-001) were purchased from Shanghai Model Organisms, Shanghai, China. At the Animal Center of Fudan University, China, mice were reared under specific pathogen-free (SPF) conditions by providing autoclaved food and water. A549-PDL1-Luc-CCL20 cells were collected, resuspended in a small amount of PBS, and washed three times with PBS. On day 0, 3 × 10^6^ cells (100 μL) were injected subcutaneously into the lateral abdomen of mice. The four groups (MOCK T, EGFR CAR-T, EGFR CAR-E27-T, and EGFR CAR-E27-CCR6-T) were randomly assigned to five mice in each group. The T cells in each group were collected, washed three times with PBS, and resuspended in a small amount of PBS.

On days 7, 10, and 13, 5 × 10^6^ T cells (100 μL) were injected into each mouse through the tail vein. Bioluminescence imaging was performed weekly to monitor the tumor changes. The injection was performed by anesthetizing the mice with 300 mg/kg of tribromoethanol injected into the abdominal cavity, followed by a tail vein injection of 150 mg/kg body weight of in vivo fluorescence imaging substrate (Promega, Madison, WI, USA). The imaging exposure was performed 3 min later to obtain an in vivo imager (Bathold Technologies, Baden-Württemberg, DE, Germany).

The exposure time of the bioluminescence intensity was set at 2 min, and the upper and lower thresholds were 65,535 and 10,000 cps, respectively. Living imaging was performed with bioanalytical instruments (Bathold Technologies, Baden-Württemberg, DE, Germany), and data were acquired using IndiGO software (Bathold Technologies, Baden-Württemberg, DE, Germany).

### 4.9. Histopathological Testing

The heart, liver, spleen, lungs, kidneys, and tumor masses were dissected. All dissected mice were sacrificed 39 days after subcutaneous tumor seeding. Tissue specimens were fixed with 4% buffered formaldehyde, and paraffin sections were analyzed using hematoxylin-eosin staining and immunohistochemistry. H&E staining of major organs was used for the preliminary safety evaluation of CAR-T cells in vivo, and immunohistochemical staining of the spleen was used to evaluate CAR-T cell persistence and infiltration. The primary and secondary antibodies used in this study were rabbit anti-human CD3 and hrp-labeled goat anti-rabbit IgG H&L (Abcam, Cambridgeshire, GB, UK), respectively, followed by DAPI staining (Abcam, Cambridgeshire, GB, UK) for nuclear staining.

### 4.10. Statistical Analysis

A Student’s *t*-test, one-way analysis of variance (ANOVA), or two-way ANOVA statistical analysis methods were chosen according to the experimental grouping. *p* < 0.05 (*) means that the *p* value should be calculated in the data with differences, and *p* < 0.01 (**) and *p* < 0.001 (***) means that the data with significant differences are to be repeated in all experiments three times.

## Figures and Tables

**Figure 1 ijms-24-05424-f001:**
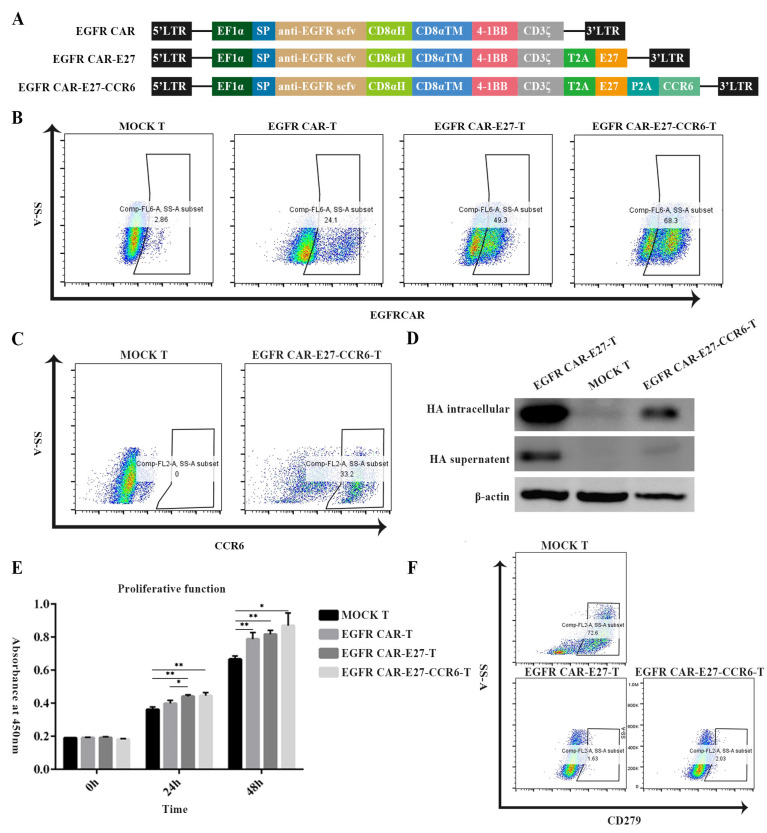
The construction and proliferation of chimeric antigen receptor (CAR) T cells. (**A**) The structure of lentiviral vector plasmids of epidermal growth factor receptor (EGFR) CAR with Programmed Death 1 (PD1) scFv E27 and CCR6. (**B**) The expression of CAR in T cells after transduction with lentiviral vector plasmids. (**C**) The expression of CCR6 after transduction with lentiviral vector plasmid EGFR CAR-E27-CCR6. (**D**) Western blot was used to detect E27 expression in transduced T cells. (**E**) The proliferation ability of the three types of CAR-T cells 24 and 48 h after stimulation compared with MOCK T. (**F**) Blocking effect of E27 on PD1 on T cells. SD is the error line, and the data difference analyzed by a unpaired *t* test is *p* < 0.05 (*), *p* < 0.01 (**).

**Figure 2 ijms-24-05424-f002:**
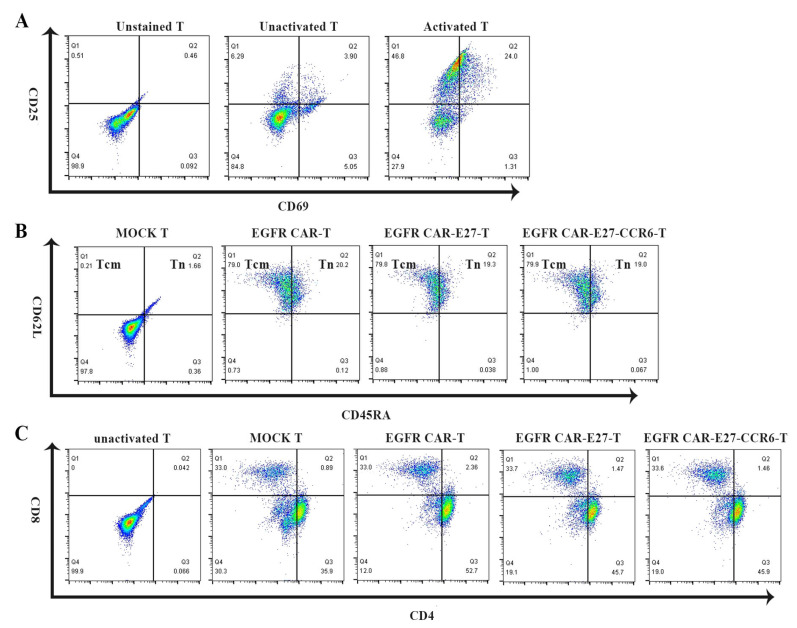
Activation phases and phenotypes of different CAR-T cells after activation and virus transduction. (**A**) Human antibodies fitc conjugated against CD25 and human pe conjugated against CD69 were mainly detected by flow cytometry, especially during the activation phase after T cell activation. (**B**) Flow cytometry was used to detect human fitc conjugate anti-CD62L antibodies and human pe conjugate anti-CD45RA antibodies. T cell functional phenotype and its ratio. (**C**) The ratio of CD4+ T cells to CD8-positive T cells was activated by viral transduction, and the human fitc anti-CD4 antibody and human pe anti-CD8 antibody were detected by flow cytometry.

**Figure 3 ijms-24-05424-f003:**
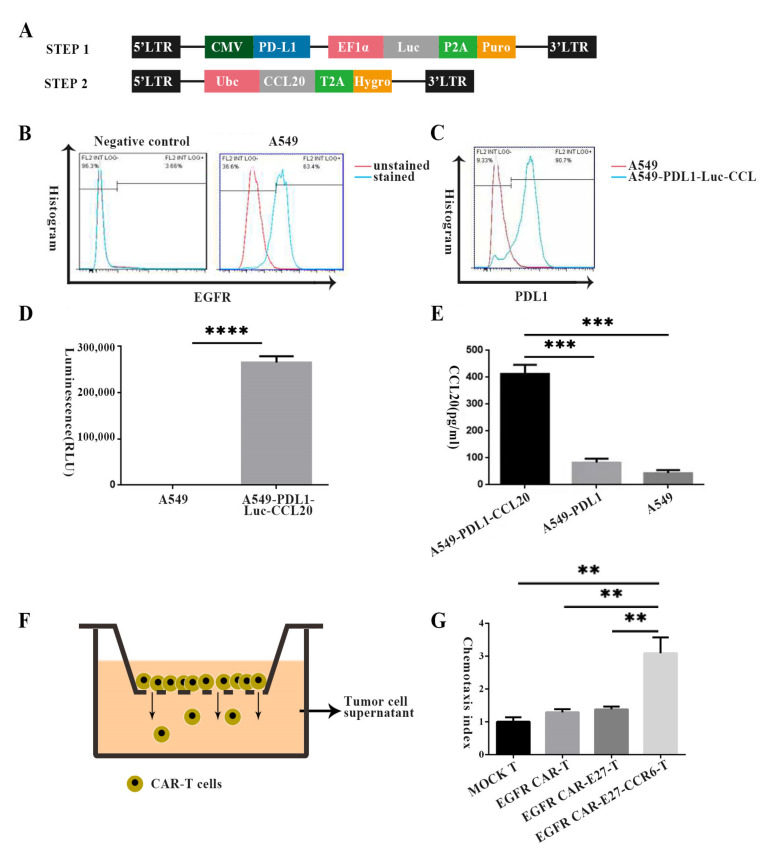
Establishment of target cells and function verification assays of CCR6 in vitro. (**A**) The structure of lentiviral vector plasmids used to transfect target cells. (**B**) EGFR expression in A549 lung cancer cells was detected by flow cytometry. (**C**) The expression of programmed death-ligand 1 (PD-L1) on target cells through flow cytometry after transfection and antibiotic screening. (**D**) The expression of Luc in target cells through luciferase detection after transfection and antibiotic screening. (**E**) The expression of CCL20 in target cells through ELISA assays after transfection and antibiotic screening. (**F**) The Transwell schematic. (**G**)The supernatant of A549-PDL1-Luc-CC20 cells induced the migration of CAR-T cells and MOCK T cells in a CCL20-dependent manner, the result of the Transwell experiment. SD represents the error line, and the statistical analysis method was the unpaired *t* test, *p* < 0.01 (**), *p* < 0.001 (***), *p* < 0.0001 (****).

**Figure 4 ijms-24-05424-f004:**
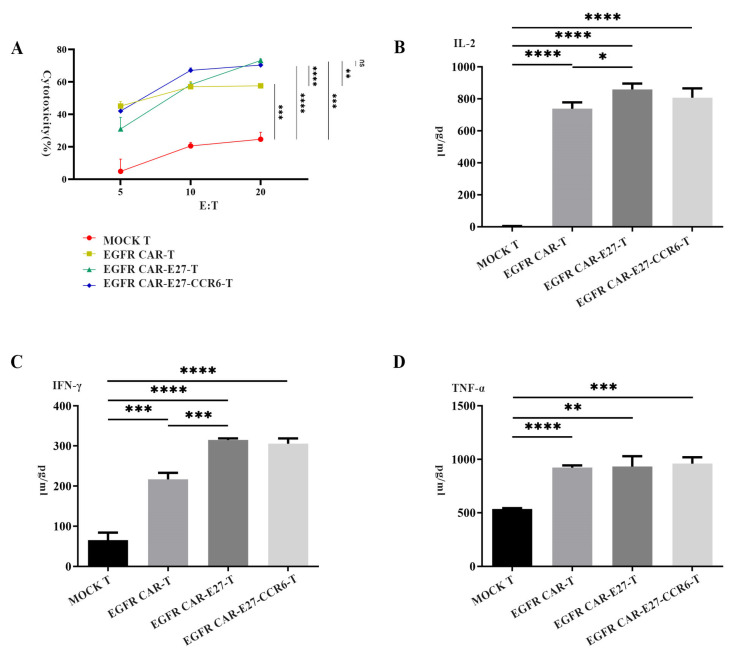
CAR-T cells specifically killed EGFR-positive tumor cells and increased cytokine release. (**A**) The killing effects of EGFR CAR-T, EGFR CAR-E27-T, EGFR CAR-E27-CCR6-T, and MOCK T in the experimental group and the control group on target cells A549-PDL2-Luc-CCL20 under different target ratios were all concluded after 6 h co-incubation; (**B**–**D**) EGFR CAR-T, EGFR CAR-E27-T, and EGFR CAR-E27-CCR6-T in the experimental group and MOCK T in the control group were co-incubated for 8 h under an E:T ratio of 10:1, and the inflammatory cytokines IL-2, IFN-γ and TNF-α secreted by the MOCK T were observed. SD is the error line representative, and the unpaired *t* test was the statistical analysis method, *p* < 0.05 (*), *p* < 0.01 (**), *p* < 0.001 (***), *p* < 0.0001 (****).

**Figure 5 ijms-24-05424-f005:**
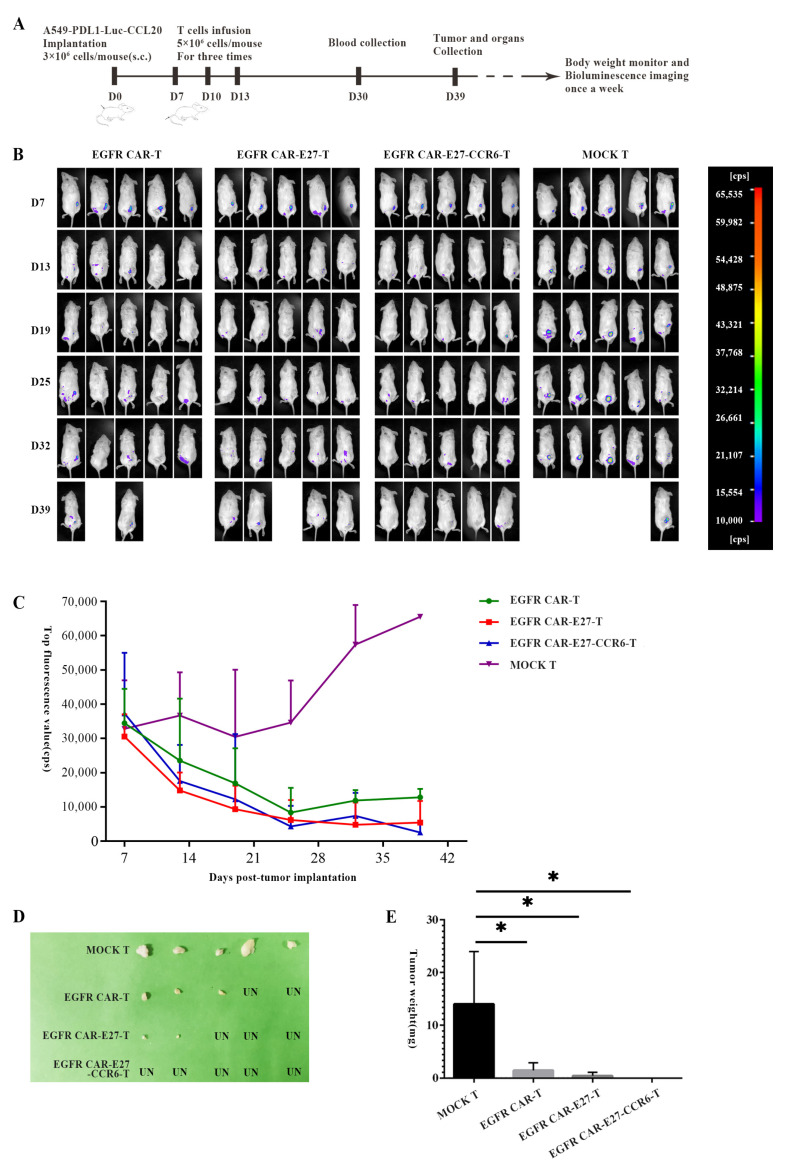
Anti-tumor function of EGFR-T cells enhanced in E27 and CCR6 in vivo. (**A**) Schematic diagram of the mouse animal model. (**B**) The monitoring chart of fluorescence values of tumor development of mice in each group at the time of D39. (**C**) Statistics on fluorescence values of mice. The highest fluorescence value was recorded at each time point. The error line represents SEM, *n* = 5. (**D**) Dissected tumors and their sizes. (**E**) Statistical results of tumor weight among different groups; SEM was the error line, and the statistical method was the unpaired *t* test, *p* < 0.05 (*).

**Figure 6 ijms-24-05424-f006:**
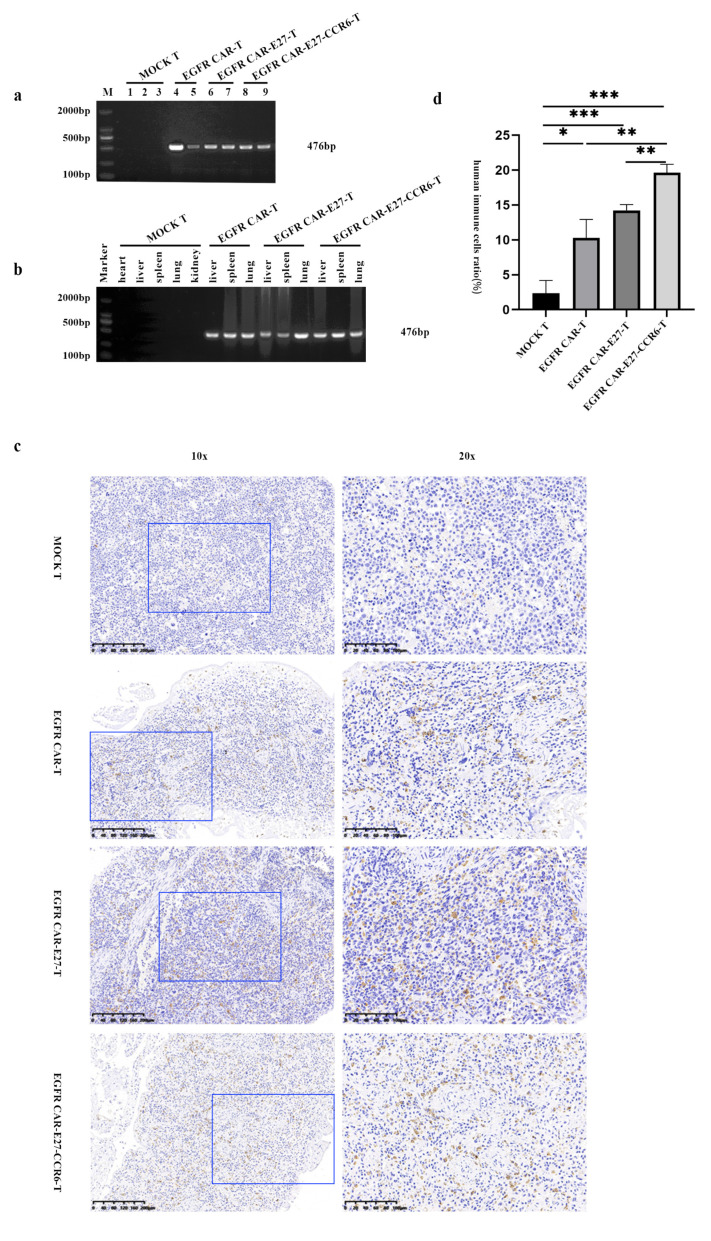
Persistence and proliferation of CAR T cells in vivo. (**a**) The exhibition of anti-EGFR scFv sequence in mouse blood genomic DNA. Lane M shows a 2000 bp DNA marker, lanes 1–3, 4–5, 6–7, and 8–9 are samples of the MOCK T, EGFR CAR-T, EGFR CAR-E27-T, and EGFR CAR-E27-CCR6-T groups, respectively. (**b**) The content of anti-EGFR scFv sequence in mouse organs genomic DNA. Lane M shows a 2000 bp DNA marker. (**c**) IHC analysis of immune cell infiltration in mice spleens. The spleen tissue paraffin sections were stained with rabbit anti-human CD3 mAb and HRP combined goat anti-rabbit IgG H & L. The brownish yellow part shows human CD3 T cells. The blue boxes in the left 10× images show the location of right 20× images. (**d**) The ratio of CD3 T cells among all mice spleen cells showed in tissue sectioning pictures of different groups. The error line shows SD, and the analysis of data differences used the unpaired *t* test, *p* < 0.05 (*), *p* < 0.01 (**), *p* < 0.001 (***).

## Data Availability

All information may be obtained from the corresponding author.

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
