# Peer review of "Chemokine Receptors CCR6 and PD1 Blocking scFv E27 Enhances Anti-EGFR CAR-T Therapeutic Efficacy in a Preclinical Model of Human Non-Small Cell Lung Carcinoma"

_ijms, 2023, doi:10.3390/ijms24065424_

Round 1
Reviewer 1 Report
Well written article
Author Response
Reviewer #1: Comments:
Point: Well written article.
Response: We thank Reviewer #1 for this encouragement.
Reviewer 2 Report
Study is well done.
Author Response
Reviewer #2: Comments:
Point: Study is well done.
Response: We thank Reviewer #2 for their encouraging comments about our article.
Reviewer 3 Report
well written paper. very interesting and topical topic
I just have a small question:
In immune checkpoint inhibitor (ICI) therapy, the microenvironment plays an important role.
what about the microenvironment and CCR6?
Author Response
Reviewer #3: Comments:
Point: well written paper. very interesting and topical topic.
I just have a small question:
In immune checkpoint inhibitor (ICI) therapy, the microenvironment plays an important role.
What about the microenvironment and CCR6?
Response: We thank Reviewer #3 for their encouraging words and for raising this pertinent point. The microenvironment of lung cancer is very important, not only when considering immune checkpoint inhibitor therapy, but also for chemokine strategy. T cells need to attach to cancer cells before killing them. However, the different locations of T cells and cancer cells in the human body make it difficult for T cells to migrate and infiltrate into tumor tissue. We reasoned that if specific properties in the tumor microenvironment facilitate T cell recruitment, it may be much easier for the T cells to find cancer cells. Thus, we used the CCL20-CCR6 axis in our study. CCL20 is highly expressed in lung cancer cells; consequently, we added CCR6 on the surface of CAR-T cells. The modified CAR-T cells to express CCR6 on the surface and they were clearly attracted to the location of lung cancer cells.
Reviewer 4 Report
Articles on the new type of cellular immunological therapy such as CART cells are always well expected. CAR-T cells have already been approved in many hematological cancers given their undeniable efficacy; however, in the field of oncology the results are more controversial.
The idea behind the work of enhancing the migration of CAR T lymphocytes by exploiting cytokines produced by the tumor is interesting; also of interest is the secretion of substances capable of enhancing killing activity against neoplastic cells.
However, the work needs to be better explained in some of its parts :
-Lines 29-35
I suggest editing the entire paragraph because it’s unclear
From line 35
I suggest adding a small introduction on what CAR-T cells are.
Lines 45-47
I suggest editing this part because it’s not clear and adding references.
From line 49
I suggest specifying what EGFR is with a brief introduction
Lines 61-63
I suggest adding a more precise explanation of the sentence
Lines 65-69
You considered CCL20 as a chemokine receptor but how can CCR6 be its receptor as reported in these lines? I suggest a more detailed definition
Lines 72 75
What is scFv E27 and what is its role?
Lines 98-107
The period is not clear.
Line 134
Please, add adequate reference
Lines 216-217
I think the sentence is incomplete.
Lines 250 251
The sentence is incomplete.
General consideration
- I think the rationale of the study should be better explained, as well as the various types of CAR cells evaluated
- Please better explain the reason for the use of checkpoint inhibitors
I await the suggested revisions
Thank you
Author Response
Reviewer #4: Comments:
Articles on the new type of cellular immunological therapy such as CART cells are always well expected. CAR-T cells have already been approved in many hematological cancers given their undeniable efficacy; however, in the field of oncology the results are more controversial.
The idea behind the work of enhancing the migration of CAR T lymphocytes by exploiting cytokines produced by the tumor is interesting; also of interest is the secretion of substances capable of enhancing killing activity against neoplastic cells.
However, the work needs to be better explained in some of its parts:
Point 1: Lines 29-35
I suggest editing the entire paragraph because it’s unclear.
Response 1: “According to the World Health Organization, in the year 2020, the incidence of lung cancer (12.2 %) was second only to that of breast cancer , making it one of the major diseases that threaten human health and life. Notably, in the same year lung cancer was the most lethal cancer, accounting for 18.2 % of all cancer-related deaths [1].
Non-small cell lung carcinoma (NSCLC) accounts for approximately 85 % of all lung cancers [2], and has a dismal 5-year survival rate of 5 % [3]. Surgery, radiotherapy, and chemotherapy are the main treatment methods for NSCLC. However, their curative effect is limited and consequently the ordinary stage (IIIA) 5-year survival rate is a meagre 15 %) [4].” (Lines 29-37 in the revised manuscript)
Point 2: From line 35
I suggest adding a small introduction on what CAR-T cells are.
Response 2: This is a valid point raised by the Reviewer. We have added a brief introduction of CAR-T in the line# with its reference. The text reads as follows:
“CARs are engineered synthetic receptors that are incorporated into the cell membranes of lymphocytes, most commonly T cells, redirecting them to recognize and eliminate cells expressing a specific target antigen. CARs function by binding to target antigens expressed on the cell surface in a process that is independent from that of MHC receptors, thereby stimulating vigorous T cell activation leading to a powerful anti-tumor response [9].” (Lines 47-52 in the revised manuscript)
The reference is as follows:
Sadelain, M., Brentjens, R., & Rivière, I. (2013). The basic principles of chimeric antigen receptor design. Cancer discovery, 3(4), 388–398.
Point 3: Lines 45-47
I suggest editing this part because it’s not clear and adding references.
Response 3: Thank you for this insightful comment. We have modified this text to make it more accurate and added references of these clinical trials:
“Clinical trials of CAR-T therapy targeting mesothelin (MSLN) [24], epidermal growth factor receptor (EGFR) [25], human epidermal growth factor receptor 2 (HER2) [26], and carcinoembryonic antigen (CEA) [27] in NSCLC have been completed in recent years but with disappointing results.” (Lines 58-61 in the revised manuscript)
Point 4: From line 49
I suggest specifying what EGFR is with a brief introduction.
Response 4: These comments were very helpful in revising and improving our manuscript. We have included a short introduction of EGFR (with reference) at line 55. It reads as follows:.
“EGFR belongs to the ERBB (Receptor protein-tyrosine kinase) family of tyrosine kinase receptors. The EGFR signaling cascade is a key regulator of cell proliferation, differentiation, division, survival, and cancer development [29].” (Lines 66-68 in the revised manuscript)
Sabbah, D. A., Hajjo, R., & Sweidan, K. (2020). Review on Epidermal Growth Factor Receptor (EGFR) Structure, Signaling Pathways, Interactions, and Recent Updates of EGFR Inhibitors. Current topics in medicinal chemistry, 20(10), 815–834.
Point 5: Lines 61-63
I suggest adding a more precise explanation of the sentence.
Response 5: We appreciate this suggestion. We have replaced this sentence, which now reads as follows:
“Another cause of unsatisfactory CAR-T cell treatment for solid tumors may be the limited migration of CAR-T cells. An efficient migration to and infiltration into tumor tissue by CAR-T cells is required to effectively kill tumor cells and eliminate tumors [42].” (Line 80-82 in the revised manuscript)
Point 6: Lines 65-69
You considered CCL20 as a chemokine receptor but how can CCR6 be its receptor as reported in these lines? I suggest a more detailed definition.
Response 6: We apologize for the confusion caused by this error. As the name suggests, chemokine ligand 20 (CCL20) is a small chemokine, and its receptor is called CC motif chemokine receptor (CCR6). The CCL20-CCR6 is a well-known ligand-receptor pair that is responsible for chemoattraction. We have corrected this error in the article.
Point 7: Lines 72-75
What is scFv E27 and what is its role?
Response 7: “scFv E27” is a variable single-chain fragment of an antibody against PD-1 We have revised and added it around the location of lines 100-101 in the revised manuscript. Brentjens Renier et al. named it “E27” in the patent WO 2016/210129 A1. This information is provided in section 4.2. of the “Materials and Methods”.
Point 8: Lines 98-107
The period is not clear.
Response 8: We have trimmed this paragraph and ensured correct and definite information is present in the text, which reads as follows:
“The initial objective of this study was to determine the proliferation rate of MOCK T, EGFR CAR-T, EGFR CAR-E27-T, and EGFR CAR-E27-CCR6-T cells after activation and lentiviral infection. The proliferation ability of T cells can be enhanced by modifying CAR and E27, as shown in the results. However, the expression of CCR6 did not significantly affect the CAR-T cell proliferation (Figure 1E).
To identify the function of E27, PD1 expression was tested at 72 h after lentiviral infection of T cells. Compared with MOCK T cells, the expression of PD1 was almost undetectable in both EGFR CAR-E27-T and EGFR CAR-E27-CCR6-T cells, confirming the blocking effect of E27 on PD1 (Figure 1F). This may be due to the single-chain antibody E27 secreted by CAR-T, which binds to the PD1 receptor on the CAR-T surface, meaning that the secretion of E27 has functional activity.” (Lines 125-135 in the revised manuscript)
Point 9: Line 134
Please, add adequate reference.
Response 9: We thank the reviewer for catching this error. The information showed in parentheses referred to the article below, we already had this reference in our reference list.
Sharma, S. V., Bell, D. W., Settleman, J., & Haber, D. A. (2007). Epidermal growth factor receptor mutations in lung cancer. Nature reviews. Cancer, 7(3), 169–181.
Point 10: Lines 216-217
I think the sentence is incomplete.
Response 10: We apologize for this error. We have revised and modified the sentence in lines 225 and 226 of the revised manuscript.
“The tumor tissues showed statistical differences between the MOCK T group and the other groups (Figure 5D).” (Lines 254-255 in the revised manuscript)
Point 11: Lines 250-251
The sentence is incomplete.
Response 11: Again, we apologize for this error and thank Reviewer #4 for catching it. We have checked the specific statement and completed the sentence in lines 259-260 of the revised manuscript.
“However, the efficacy of CAR-T cells in treating solid tumors is limited [50].” (Line 291 in the revised manuscript)
Point 12: General consideration
- I think the rationale of the study should be better explained, as well as the various types of CAR cells evaluated.
- Please better explain the reason for the use of checkpoint inhibitors.
Response 12: We thank Reviewer for these suggestions. Our manuscript has benefitted from the content of all comments. Based on these comments, we have revised our article, especially the “Introduction”. And here is the reason why we start this study using CAR-T with checkpoint inhibitors.
In recent years, CAR therapy has become the most potent therapy for cancer treatment. However, till now, the issue of immune escape, mediated by immune checkpoint, and the problem of migration and infiltration of T cells obstructed the use of CAR-T therapy against solid tumors. To improve the treatment for NSCLC, we selected the checkpoint inhibitor strategy and chemokine strategy to modify CAR-T therapy, hoping to solve the above-mentioned two issues.
Reviewer 5 Report
In my opinion, the article is well written, the topic is definitely important, the methods used are definitely modern and adequate to the problem. The conclusions are consistent with the results obtained. The article may be published as presented.
Author Response
Reviewer #5: Comments (Round 1):
Point: In my opinion, the article is well written, the topic is definitely important, the methods used are definitely modern and adequate to the problem. The conclusions are consistent with the results obtained. The article may be published as presented.
Response: We thank Reviewer #5 for their encouraging comments about our article.
Reviewer 6 Report
In this work the authors confirmed the anti-tumor function of EGFR 23 CAR-T cells by PD1 blocking and CCR6 expression using an NSCLC xenograft model. These finding implement the knowledge on efficacy of CAR-T as treatment strategy for NSCLC cure. There are minor revisions that I report below. In my opinion the work can accept with minor revisions.
Minor revisions:
1. More description regarding the generation of CAR- T cells used in the study.
2. Line 14: (,) I assume was added by error.
3. Lines 87-88: CCL20 was written twice.
4. Line 112: (EGFR CAR-E27 -T cells) is not written.
5. Line 165: I assume PCDH should be written pCDH.
Author Response
Reviewer #6: Comments (Round 1):
In this work the authors confirmed the anti-tumor function of EGFR 23 CAR-T cells by PD1 blocking and CCR6 expression using an NSCLC xenograft model. These finding implement the knowledge on efficacy of CAR-T as treatment strategy for NSCLC cure. There are minor revisions that I report below. In my opinion the work can accept with minor revisions.
Minor revisions:
Point 1: More description regarding the generation of CAR- T cells used in the study.
Response 1: This is a valid point raised by the Reviewer. We have added more details of the generation of our CAR-T cells in the study, as follows.
“To obtain EGFR CAR-T, EGFR CAR-E27-T, and EGFR CAR-E27-CCR6-T cells, the DNA sequence of anti-EGFR scFv, E27 and CCR6 was collected from former articles [55], and lentiviral vectors for EGFR CAR, EGFR CAR-E27, and EGFR CAR-E27-CCR6 were constructed as described in the Materials and Method (Figure 1A). The cells from CD3+T were isolated from human peripheral blood mononuclear cells (PBMCs) of healthy donors using isolation kit and infected with lentivirus after 72 h of activation.”
(Lines 106-111 in the revised manuscript)
Point 2: Line 14: (,) I assume was added by error.
Response 2: Thank you for this insightful comment. We have modified this text and deleted the wrong (,).
“T cell was engineered to express the chemokine receptor CCR6 and secrete PD1, blocking scFv E27 to enhance their anti-tumor effects.”
(Line 14 in the revised manuscript)
Point 3: Lines 87-88: CCL20 was written twice.
Response 3: We apologize for the confusion caused by this error. We have deleted the repetitive in the taxt.
“Chemokines CCL19, CXCL13, and CCL20 are chemokines with highly expression [44,45] in lung adenocarcinoma, a conclusion of The Cancer Genome Atlas (TCGA) project initiated by the National Cancer Institute in the United States.”
(Lines 87-90 in the revised manuscript)
Point 4: Line 112: (EGFR CAR-E27 -T cells) is not written.
Response 4: We thank the reviewer for catching this error. We have added the missing word into the text.
“The activation, phenotype, and expression of CAR- and CCR6-T cells were detected, and the results showed that 24.1, 49.3, and 68.3 % of them (Figure 1B) expressed an-ti-EGFR scFv on EGFR CAR-T, EGFR CAR-E27-T and EGFR CAR-E27-CCR6-T cells, and 33.2 % expressed CCR6 in EGFR-CAR-E27-CCR6-T cells (Figure 1C).”
(Lines 111-115 in the revised manuscript)
Point 5: Line 165: I assume PCDH should be written pCDH.
Response 5: We thank Reviewer for these suggestions. We have changed the proper noun used in the text.
“A549 cells were transduced with pPCDH-PDL1-Luc-puro lentivirus, followed by puromycin selection.”
(Line 167-168 in the revised manuscript)
Round 2
Reviewer 4 Report
Thank you for the revisione. It's ok for me
Author Response
Reviewer #4: Comments (Round 2):
Point: Thank you for the revision. It's ok for me.
Response: We thank Reviewer #4 for this encouragement.